

# A global-local neighborhood search algorithm and tabu search for flexible job shop scheduling problem

Nayeli Jazmin Escamilla Serna, Juan Carlos Seck-Tuoh-Mora,
Joselito Medina-Marin, Norberto Hernandez-Romero,
Irving Barragan-Vite and Jose Ramon Corona Armenta

AAIyA-ICBI-UAEH., Mineral de la Reforma, Hidalgo, Mexico

## ABSTRACT

The Flexible Job Shop Scheduling Problem (FJSP) is a combinatorial problem that continues to be studied extensively due to its practical implications in manufacturing systems and emerging new variants, in order to model and optimize more complex situations that reflect the current needs of the industry better. This work presents a new metaheuristic algorithm called the global-local neighborhood search algorithm (GLNSA), in which the neighborhood concepts of a cellular automaton are used, so that a set of leading solutions called smart-cells generates and shares information that helps to optimize instances of the FJSP. The GLNSA algorithm is accompanied by a tabu search that implements a simplified version of the Nopt1 neighborhood defined in *Mastrolilli & Gambardella (2000)* to complement the optimization task. The experiments carried out show a satisfactory performance of the proposed algorithm, compared with other results published in recent algorithms, using four benchmark sets and 101 test problems.

## INTRODUCTION

The scheduling of jobs and resource assignments in a production system includes a series of combinatorial problems that continue to be widely investigated today to propose and test new metaheuristic algorithms.

One of these problems is the Flexible Job Shop Scheduling Problem (FJSP), an extension of the Job Shop Scheduling Problem (JSP). This problem consists of assigning a set of jobs to be processed on multiple machines. Each job consists of several operations that must be processed sequentially. Operations of different jobs can be interlarded in the scheduling. The system's flexibility is given with the possibility that each operation (perhaps all of them) of a given set can be processed in several machines.

Thus, the FJSP aims to find the best possible machine assignment and the best possible scheduling of operations; classically, the objective is to find the shortest possible time (or makespan) to process all the jobs.

The FJSP continues to be a very active subject of research, since its first definition in *Brucker & Schlie (1990)*. Many of the latest works have focused on presenting hybrid

Corresponding author
Juan Carlos Seck-Tuoh-Mora,
jseck@uaeh.edu.mx

techniques for better results. For example, an algorithm has been proposed in which genetic operators and the tabu search (TS) interact (*Li & Gao, 2016*). Another work combines discrete bee colony operators with the TS to optimize classic problems and problems with job cancellation and machine breakdowns (*Li et al., 2017*). Other recent work uses discrete particle swarm operators with hill climbing and random restart to optimize well-known test problems (*Kato, de Aguiar Aranha & Tsunaki, 2018*). These references are just a small sample of the works that develop hybrid techniques of discrete operators and local searches to propose new algorithms that improve the makespan's calculation in FJSP instances.

Within the metaheuristic algorithms, a new optimization strategy has been proposed using concepts of cellular automata. Relevant work is proposed in *Shi et al. (2011)*, where different types of cellular automaton-like neighborhoods are used in conjunction with particle swarm operations for continuous global optimization.

Other related work using cellular automaton-like neighborhoods to design IIR filters can be consulted in *Lagos-Eulogio et al. (2017)*. Recently, another discrete optimization algorithm has been presented for the concurrent layout scheduling problem in the job-shop environment (*Hernández-Gress et al., 2020*), where the idea of cellular automata inspires the neighborhood strategy used by the proposed algorithm.

The idea that a solution can be improved by developing a neighborhood with new solutions generated by small changes in its current information and by sharing information with other solutions is the inspiration behind these algorithms, like the evolution rule of a cellular automaton (*McIntosh, 2009*).

Following this trend developed in previous studies, this work proposes that the application of this neighborhood idea will optimize instances of the FJSP. Specifically, an algorithm is proposed that uses a set of leading solutions called smart-cells. In each iteration of the algorithm, the population of smart-cells is selected using elitism and tournament selection.

With this selected population, each smart-cell generates a neighborhood of new solutions using classical operators of combinatorial problems (insertion, swapping, and path relinking (PR)). The best one is selected from this neighborhood, which updates the smart-cell value.

The neighborhood-based optimization of each smart-cell is complemented by a TS using a simplified version of the Nopt1 neighborhood proposed in *Mastrolilli & Gambardella (2000)*. In this neighborhood, a random critical path is selected, and a better solution is sought, perhaps a machine minimizing the makespan for each operation on the critical path but without changing its position in the current array of operations.

This neighborhood management of each smart-cell and the simplified neighborhood Nopt1 allows us to obtain an algorithm of less complexity than those previously proposed, which can adequately solve the test problems commonly used in the specialized literature. In particular, for instances of FJSP with high flexibility (where more machines can perform the same operation), two solutions are presented with better makespan values compared to the algorithms reported in this manuscript.

The structure of the paper is as follows. "State of the art of FJSP" presents a state of the art of the FJSP. "Problem formulation" describes the FJSP formulation. "Global-local neighborhood search algorithm for the FJSP" explains the strategy and operators used for global and local searches that define the global-local neighborhood search algorithm (GLNSA). "Experimental results" shows the experimental results obtained when the GLNSA is applied to instances of the FJSP with high flexibility. The last section provides the conclusions of the article and prospects for future work.

## STATE OF THE ART OF FJSP

As mentioned, the FJSP is an extension of the classic JSP. The classic problem seeks to find the assignment of operations in a set of predefined machines, while the flexible case consists of a sequence of operations, where each operation can be performed on several available machines, possibly with different processing times. To solve an instance of the FJSP, one must consider two sub-problems: assignment and scheduling (*Brandimarte, 1993*). For each operation, the first assigns a machine from a set of available ones. The second is in charge of sequencing the operations assigned to each machine to obtain a feasible schedule to minimize the objective function (*Li, Pan & Tasgetiren, 2014*).

The problem definition was introduced by *Brucker & Schlie (1990)*, who proposed a polynomial-graphical algorithm to solve a problem with only two jobs, concluding that FJSP belongs to the category of NP-hard problems for which there are no algorithms that can bring optimal solutions in polynomial time.

One of the first works to address the FJSP with a heuristic approach is *Brandimarte (1993)*, which uses dispatch rules and a hierarchical TS algorithm to solve the problem and introduce 15 instances.

Since then, many investigations have addressed the FJSP and applied different approaches and methods to solve it. For example, *Mastrolilli & Gambardella (2000)* introduces two neighborhood functions to use local search techniques by proposing a TS procedure.

A practical hierarchical solution approach is proposed in *Xia & Wu (2005)* to solve multiple targets for the FJSP. The proposed approach uses particle swarm optimization (PSO) to assign operations in machines and the simulated annealing (SA) algorithm to each machine's program operations. The objective is to minimize the makespan (maximum completion time), the total machine workload, and the critical machine workload.

A genetic algorithm (GA) to be applied to the FJSP is proposed in *Pezzella, Morganti & Ciaschetti (2008)*. The developed algorithms integrate different selection and reproduction strategies and show that an efficient algorithm is developed when different rules to find the initial population, selection, and reproduction operators are combined. Another hybridized GA (HGA) is described in *Gao, Sun & Gen (2008)*, which strengthens the search for individuals and is improved with the variable neighborhood descent (VND) variable; since it is a multi-objective problem, HGA seeks the minimum makespan, maximum workload, and minimum total workload. Two local search procedures are used, the first for a moving operation and the second for two moving operations.

Hybridization of two algorithms, PSO and TS, are combined in *Zhang et al. (2009)* to solve a multi-objective problem, that is, several conflicting objectives, mainly in large-scale problems, where the PSO has a high search efficiency combining local and global searches and TS is used to find a near-optimal solution.

In *Amiri et al. (2010)*, a variable neighborhood search (VNS) algorithm applied to the FJSP is proposed, and its objective function is the makespan. Several types of neighborhoods are presented, where assignment and sequence problems are used to generate neighboring solutions. Another hybrid algorithm (HA) using TS and VNS is presented in *Li, Pan & Liang (2010)*, which considers three minimization objectives, produces neighboring solutions in the machine assignment module, and performs local searches in the operation scheduling.

An algorithm that considers parallel machines and maintenance costs in the FJSP is exposed in *Dalfard & Mohammadi (2012)*, which proposes a new mathematical model that applies the HGA and the SA algorithm, obtaining satisfactory results in 12 experiments using multiple jobs.

The work in *Yuan, Xu & Yang (2013)* adapts the harmony search algorithm (HS) in the FJSP. They developed techniques to convert the continuous harmony vector into two vectors, and these vectors are decoded to reduce the search space applied to an FJSP. Finally, they introduce an initialization scheme by combining heuristic and random techniques and incorporating the local search in the HS, in order to speed up the local search process in the neighborhood.

Another discrete algorithm based on an artificial-bee colony, called DABC, is presented in *Li, Pan & Tasgetiren (2014)*. They take three objectives as their criteria, where they adopt a self-adaptive strategy, represented by two discrete vectors and a TS, demonstrating that its algorithm is efficient and effective with high performance.

A GA that incorporates the Taguchi method in its coding to increase its effectiveness is exposed in *Chang et al. (2015)* and it evaluates the performance of the proposed algorithm using the results of *Brandimarte (1993)*. A hybrid evolutionary algorithm based on the PSO and the Bayesian optimization algorithm (BOA) is developed in *Sun et al. (2015)* and used to determine the relationship between the variables and its objective to minimize the processing time and improve the solutions and robustness of the process.

A multi-objective methodology is described in *Ahmadi et al. (2016)* for the FJSP in the specific case of machine breakdown situations. They use two algorithms, the non-dominated sorting genetic algorithm (NSGA) and the NSGA-II, which is usually utilized to solve large multi-objective problems, like evaluating the status and condition of machine breakdowns. A HA that uses a GA and a TS to minimize the makespan is presented in *Li & Gao (2016)*. The proposed algorithm has an adequate search capacity and balances intensification and diversification very well.

In *Li et al. (2017)*, the hybrid artificial bee colony (HABC) algorithm and the improved TS algorithm are proposed to solve the FJSP in a textile machine company. Three rescheduling strategies are introduced—schedule reassembly, schedule intersection, and schedule insertion—to address dynamic events such as new jobs inserted, old jobs, and when there may be cell and machine breakdowns. The HABC algorithm is shown to have

satisfactory exploitation, exploration, and performance to solve the FJSP. A non-dominant genetic classification algorithm that serves as an evolutionary guide for an artificial bee colony (BEG-NSGA-II) is developed in *Deng et al. (2017)*, and it focuses on the multi-objective problem (MO-FJSP). Usually, this type of algorithm converges prematurely. Therefore, that paper uses a two-stage optimization to avoid these disadvantages in order to minimize the maximum completion time, the workload of the most loaded machine, and the total workload of all the machines. An optimization algorithm applying a hybrid ant colony optimization (ACO) to solve the FJSP described in *Wu, Wu & Wang (2017)* is based on a 3D disjunctive graph, and has four objectives: to minimize the completion time, the delay or anticipation penalty cost, average machine downtime, and the cost of production.

In *Shen, Dauzère-Pérès & Neufeld (2018)* the FJSP is addressed using sequence-dependent setup time (SDST) and a mixed-integer linear programming model (MILP) to minimize the makespan using the TS as an optimization algorithm. They apply specific functions and a diversification structure, comparing their model with well-known reference instances and two metaheuristics from the literature, obtaining satisfactory results. In *Kato, de Aguiar Aranha & Tsunaki (2018)*, new strategies are used in population initialization, particle displacement, stochastic assignment of operations, and partially and fully flexible scenario management, to implement a HA using the PSO for the machine routing subproblem and explore the solution space with a Random Restart Hill Climbing (RRHC) for the local search programming subproblem. A new definition of the FJSP (double flexible job-shop scheduling problem, DFJSP) is described in *Gong et al. (2018)*. Here, the processing time was considered, and factors related to the environment's protection were presented as an indicator. They presented and resolved ten benchmarks using the new algorithm. An algorithm that combines the uncertainty processing time to solve an FJSP in order to minimize uncertain times and the makespan is presented in *Xie & Chen (2018)*. The algorithm uses gray information based on external memory with an elitism strategy. In *Reddy et al. (2018)*, the FJSP is solved for the minimization of the makespan and the workload of the machines using a programming model (mixed-integer non-linear programming, MINLP) with machines focused on real-time situations using a new HA through PSO and GA to solve multiple objectives, obtaining high-quality solutions. An algorithm that addresses the FJSP to minimize the total workflow and inventory costs is described in *Meng, Pan & Sang (2018)*; it applies an artificial bee colony (ABC) and the modified migratory bird algorithm (MMBO), to obtain a satisfactory capacity search. In *Nouiri et al. (2018)*, 13 benchmark test problems of the FJSP are taken to prove the effectiveness of a distributed particle swarm optimization algorithm.

In *Tang et al. (2019)*, two optimization methods are applied with two significant characteristics in practical casting production: the Tolerated Time interval (TTI) and the Limited Start Time interval (LimSTI). A model to calculate the energy consumption of machinery is presented in *Wu, Shen & Li (2019)*, which has different states and a deterioration effect to determine the real processing time to apply a hybrid optimization using the SA algorithm. A new algorithm called hybrid multi-verse optimization (HMVO) is proposed in *Lin, Zhu & Wang (2019)* to treat a fuzzy problem in an FJSP. Route linking

technique is used, and a mixed push-based phase to expand the search space and local search to improve the solution is incorporated. Another elitist non-dominated sorting HA (ENSHA) for a MO-FJSP is explained in *Li et al. (2019)*. A configuration dependent on the sequence is used. Its objective is to find the minimum makespan and the total costs of the installation, by proposing a learning strategy based on estimating the distribution algorithm (EDA) and checking its effectiveness with 39 instances and a real case study. Another HA based on genetic operators is presented in *Huang & Yang (2019)*, and applied to MO-FJSP, considering transportation time. TS continues to be a standard method to complement the local search in the solution of FJSP, as demonstrated in *Kefalas et al. (2019)*, which proposes a memetic algorithm for the multi-objective case. Hybridization is continuously applied to propose new and improved algorithms for the FJSP (*Gao et al., 2019*). One proposal is in *Zarrouk, Bennour & Jemai (2019)*, where a two-level PSO algorithm is presented and tested with 16 benchmark problems. Another GA with a VNS is explained in *Zhang et al. (2019)* and proved with 13 well-known benchmark problems. Another modified metaheuristic is presented in *Luan et al. (2019)*, where the whale algorithm is adapted for the FJSP and proved with 15 test problems. A HA that combines PSO and TS is described in *Toshev (2019)* and the performance of the algorithm is analyzed in 12 benchmark problems. A distributed approach for implementing a PSO method is explained in *Caldeira & Gnanavelbabu (2019)* and proved with 6 different benchmark datasets.

In *Chen et al. (2020)*, 14 benchmark test problems are used to demonstrate the efficiency of a self-learning GA based on reinforcement learning in the FJSP. A fuzzy version of the FJSP is studied in *Vela et al. (2020)*, where an evolutionary algorithm is proposed, using a TS again for optimizing a due-date cost. Dynamic flexibility in FJSP is analyzed in *Baykasoğlu, Madenoğlu & Hamzaday (2020)* with a greedy randomized adaptive search. The efficiency of the proposed algorithm is proved with three different sets of benchmark test problems. Other hybrid methods for optimizing the FJSP are described in *Bharti & Jain (2020)*, where 28 benchmark instances are taken from three different data sets to demonstrate the performance of these methods. In *Shi et al. (2020)*, a multi-population genetic algorithm with ER network is proposed and tested with 18 benchmark problems. MO-FJSP instances are solved with a hybrid non-dominated sorting biogeography-based optimization algorithm (*An et al., 2021*).

Previous works show that the algorithms dedicated to solving FJSP instances that have had the best results use hybrid techniques that combine metaheuristic techniques and local search methods such as TS. Another point analyzed in the literature review is that the most recurrent objective function is the makespan as the most widely used performance measure.

## PROBLEM FORMULATION

The FJSP is presented following the definition of *Zuo, Gong & Jiao (2017)*. There is a set of $n$ jobs $J = \{J_1, J_2, \ldots J_n\}$ and a set of $m$ machines $M = \{M_1, M_2, \ldots M_m\}$. Each $J_i$ job consists of a sequence of operations $O J_i = \{O_{i, 1}, O_{i, 2}, \ldots, O_{i, n_i}\}$, where $n_i$ is the number of operations contemplated by the job $J_i$. For $1 \leq i \leq n$ and $1 \leq j \leq n_i$, each operation $O_{i, j}$ can

be processed by one machine from a set of machines $M_{i,j} \subseteq M$. The processing time of $O_{i,j}$ on the machine $M_k$ is denoted by $p_{i,j,k}$.

In an instance of the FJSP, the following conditions are considered:

1. An operation cannot be interrupted while a machine is processing it.

2. One machine can process one operation at most.

3. Once the order of operations has been determined, it cannot be modified.

4. Breakdowns in machines are not considered.

5. The works are independent of one another.

6. The machines are independent of one another.

7. The time used for the machines' preparation and the transfer of operations between them is negligible.

A solution for the FJSP is defined as the order of operations $O_{i,j}$ that respects each job's precedence restrictions. For each operation $O_{i,j}$, a machine is selected from the subset $M_{i,j}$. The objective is to find the feasible order of operations $O_{i,j}$ and for each operation, the assignment of a machine in $M_{i,j}$ minimizing the makespan, or the time needed to complete all jobs. The makespan can be formally defined as $C_{max} = max\{C_i\}$, where $C_i$ represents the completion time for all operations of the job $J_i$, for $1 \le i \le n$. A mathematical formulation of the problem can be represented by Eq. (1), with the corresponding constraints.

Objective:

$$min\{C_{max}\} = min\{max \sum_{i=1}^{n} \sum_{j=1}^{m} (s_{i,j,k} + p_{i,j,k})\} \tag{1}$$

Subject to:

$$p_{i,j,k} > 0, \; 1 \le i \le n; 1 \le j \le n_i; 1 \le k \le m \text{ such that } M_k \in M_{i,j} \tag{2}$$

$$s_{i,j,k} + p_{i,j,k} \le s_{i,j+1,k} \tag{3}$$

$$X_{i,j,k} = \begin{cases} 1, & \text{when operation} O_{i,j} \text{is processed on} M_k \\ 0, & \text{otherwise} \end{cases} \tag{4}$$

$$\sum_{k=1}^{m} X_{i,j,k} = 1 \tag{5}$$

$$\sum_{i=1}^{n} \sum_{j=1}^{n_i} X_{i,j,k} = 1 \tag{6}$$

The objective function in Eq. (1) minimizes the makespan or the maximum completion time of all jobs $J_i$, where $s_{i,j,k}$ is the start time of operation $O_{i,j}$ in machine $M_k$. Constraint in Eq. (2) represents the processing time of every operation being greater than 0. Constraint in Eq. (3) assures the precedence between operations of the same job. A processing record of the assignment of one operation to a feasible machine is represented by $X$ in Eq. (4). With $X$, constraint in Eq. (5) shows that each operation is assigned to only one machine, and constraint in Eq. (6) ensures each machine can process only one operation at any time.

Thus, an instance of the FJSP involves two problems: the scheduling of operations and the machine assignment to each operation.

A recent strategy for solving this type of problem is the hybridization of techniques that optimize both problems. This manuscript follows this research line, proposing an algorithm that combines the generation of new solutions using a neighborhood inspired by cellular automata, complemented with a local search technique based on TS, using a simplified version of the Nopt1 neighborhood presented in *Mastrolilli & Gambardella (2000)*.

This type of neighborhood allows the exploration of new solutions using well-known operators for the scheduling problem. The conjunction with the simplified neighborhood Nopt1 allows the proposal of a neighborhood-based algorithm that performs global and local searches in each iteration, with a complexity similar to the most recent algorithms, and uses less computational time, obtaining satisfactory results for problems with high flexibility.

# GLOBAL-LOCAL NEIGHBORHOOD SEARCH ALGORITHM FOR THE FJSP

## General description of the GLNSA

Many recent works have proposed hybrid optimization algorithms that combine a population method with another local search method. These methods have produced satisfactory results, and new proposals continue to be developed to have equally efficient algorithms, with a simple implementation and lower complexity of execution. In these methods, the population part applies a series of information sharing and mutation operators to improve the optimization process. Usually, these methods are used serially to each individual to obtain a new position.

In the algorithm proposed in this work, a different strategy is taken, inspired by the operation of a cellular automaton, where the solution of a complex problem is achieved through simple operators' cooperation. This inspiration has been proved before in the design of infinite-impulse filters (*Lagos-Eulogio et al., 2017*) and a concurrent layout and scheduling problem (*Hernández-Gress et al., 2020*), obtaining good results.

A cellular automaton is a discrete dynamic system made up of indivisible elements called cells, where each cell changes its state over time. The state change can depend on both the current state of each cell and its neighboring cells. With such simple dynamics, cellular automata can create periodic, chaotic, or complex global behavior (*Wolfram, 2002*; *McIntosh, 2009*; *Adamatzky, 2010*; *Bilan, Bilan & Motornyuk, 2020*).

In this work, the idea of a cellular-automaton neighborhood is an inspiration to propose a new algorithm that optimizes instances of the FJSP. The algorithm has $S_n$ leading solutions or smart-cells, where each of them first performs a global search mainly focused on making modifications to the sequencing of operations, applying several operators to form a neighborhood, and selecting the best modification. Then, each smart-cell executes a local search focused on machine assignment, applying a TS with increasing iterations at each step of the algorithm. Thus, in each iteration, a global exploration search and a local

exploitation search are performed to optimize instances of the FJSP, and hence, this process is called the global-local neighborhood search algorithm (GLNSA).

The novel part of the GLNSA is that instead of serially applying the sharing-information and mutation operators to each smart-cell to obtain a new solution as most recently proposed hybrid methods do, in the GLNSA, a neighborhood of new solutions is generated. Each neighbor is the product of applying a random information exchange or mutation operator to the original smart-cell. Then the best neighbor is selected to replace it, following a neighborhood structure inspired by cellular automata.

In this way, the contribution of GLNSA is to show that notions of cellular automata are helpful to inspire new forms of hybridization in the conceptualization of new optimization methods for the FJSP, obtaining satisfactory results as shown in the subsequent sections. The corresponding flow chart is described in Fig. 1.

The detailed explanation of the encoding and decoding of solutions, the neighborhood used, its operators for the global search, and the TS operators are presented in the following sub-sections. The general procedure of the GLNSA is in Algorithm 1.

## Encoding and decoding of solutions

To represent a solution for an instance of the FJSP, we take the encoding with two strings (*OS* and *MS*) described in *Li & Gao (2016)*, *OS* for operations and *MS* for machines.

The string *OS* consists of a permutation with repetitions where each job $J_i$ appears $n_i$ times. The string *OS* is read from left to right, and the $j-th$ appearance of $J_i$ indicates that the $O_{i,j}$ operation of job $J_i$ should be processed. This coding of the sequence of operations *OS* has the advantage of any permutation with repetitions producing a valid sequence so that the operators used in this work will always yield a feasible sequence.

The string *MS* consists of a string with a length equal to the number of the total operations. The string is divided into $n$ parts, where the $i-th$ part contains the machines assigned to $J_i$ with $n_i$ elements. For each $i-th$ part of *MS*, the $j-th$ element indicates the machine assigned to $O_{i,j}$.

Initially, each solution *OS* is generated at random, making sure that each job $J_i$ appears exactly $n_i$ times. For every operation, one of the possible machines that can process it is selected at random as well.

For each pair of *OS* and *MS* strings that represents a solution for an instance of the FJSP, their decoding is done using an active scheduling. For each operation $O_{i,j}$ in *OS* and its assigned machine $k$ in *MS*, its initial time $s(O_{i,j})$ is taken as the greater time between the completion time of the previous operation $O_{i,j-1}$ and the lesser time available in the machine $k$ (not necessarily after the last operation programmed in that machine) where there is an available time slot, such that the processing time $p_{i,j,k}$ is less than or equal to the size of this slot. The time the $O_{i,j}$ operation is completed is called $C(O_{ij})$, and for $j = 1$, $s(O_{i,j} - 1) = 0$ for all $1 \leq i \leq n$. An example with 3 jobs and 2 machines is shown in Fig. 2.

In Fig. 2 (A), there are 3 jobs, each with 2 operations. Almost every operation can be executed on the 2 available machines. On average, each operation can be executed by 1.83 machines; this is called the system's flexibility. Note that the time at which each operation is performed on each machine may be different. In part (B), you can see how a

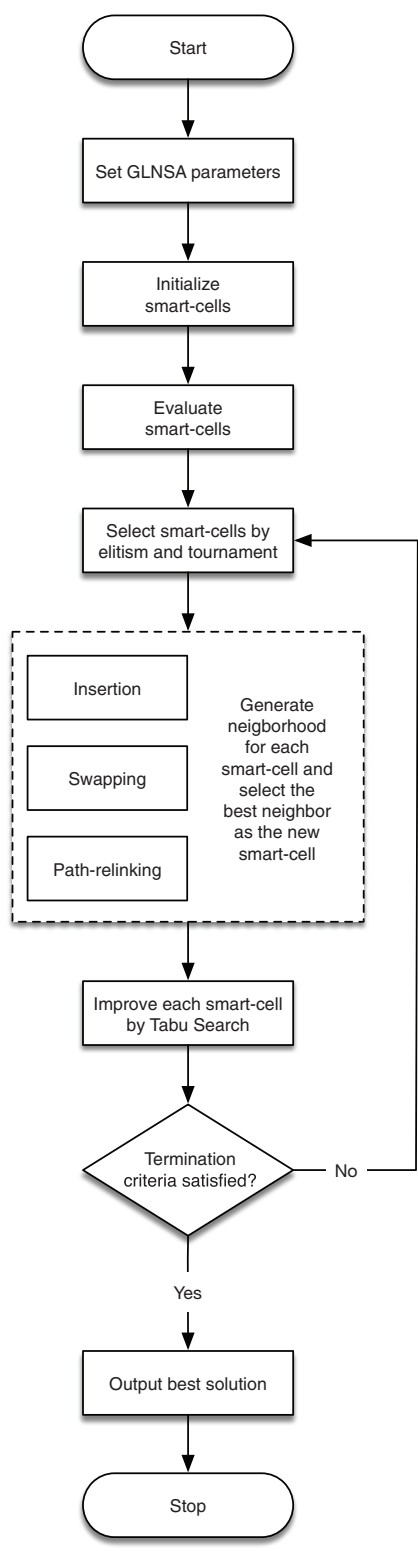

**Figure 1 The workflow of the GLNSA.**

| Algorithm 1 | General description of the GLNSA. |
|---|---|

**Result**: Best smart-cell

Set the parameters of the GLNSA;

Initialize the population of smart-cells with Sn solutions generated at random;

Evaluate each smart-cell to obtain its makespan;

**do**

> Select a refined population from the best smart-cells using elitism and tournament;
>
> For each smart-cell, generate a neighborhood (using insertion, swapping, and PR operators) and
>
> take the best neighbor as a new smart-cell;
>
> For each smart-cell, improve the machine assignment by TS;

**while** *(Iteration number less than $G_n$ or stagnation number less than $S_b$)*;

Return the smart-cell with minimum makespan;

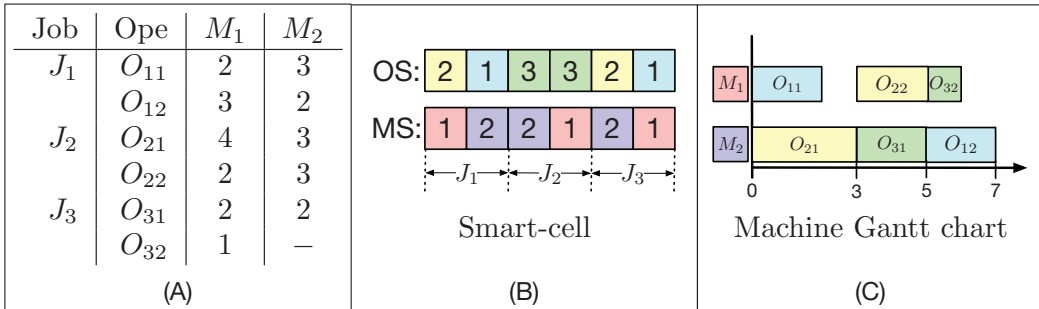

| Job | Ope | $M_1$ | $M_2$ |
|---|---|---|---|
| $J_1$ | $O_{11}$ | 2 | 3 |
|  | $O_{12}$ | 3 | 2 |
| $J_2$ | $O_{21}$ | 4 | 3 |
|  | $O_{22}$ | 2 | 3 |
| $J_3$ | $O_{31}$ | 2 | 2 |
|  | $O_{32}$ | 1 | – |

(A)  (B)  (C)

**Figure 2 Example of a FJSP instance (A), the encoding of a factible solution (B) and the active decoding (C) used by the GLNSA.**

smart-cell is encoded. It consists of two strings; the first *OS* is a permutation with repetitions, where each job appears 2 times. The second string *MS* contains the machines programmed for each operation, where the first 2 elements correspond to the machines assigned to the operations $O_{1,1}$ and $O_{1,2}$, the second block of 2 elements specify the machines assigned to operations $O_{2,1}$ and $O_2$, and so on. Finally, part (C) indicates the decoding of the smart-cell reading of the string *OS* from left to right. In this case, the $O_{22}$ operation, which is the fifth task programmed in *OS*, is actively accommodated, since there is a gap in the $M_1$ machine between $O_{11}$ and $O_{32}$, which is of sufficient length to place $O_{22}$ just after the preceding operation on the machine $M_2$ and without moving the operations already programmed in $M_1$. This scheduling gives a final makespan of 7 time units using the active decoding.

## Selection method

The GLNSA uses elitism and tournament to refine the population, by considering the value of the makespan of each smart-cell. For elitism, a proportion $E_p$ of smart-cells is taken with the best values of population's makespan. Those solutions will remain unchanged in the next generation of the algorithm. The rest of the population members are selected

using a tournament scheme, where a group of $b$ smart-cells are randomly selected from the current population and competed with each other, and the smart-cell with the best makespan is selected to become part of the population to be improved using the global and local search operators described later. In this work, we take $b = 2$. This mixture of elitism and tournament allows a balance between exploring and exploiting the information in the population, keeping the best smart-cells, and allowing the other smart-cells with a good makespan to continue in the optimization process.

## Neighborhood structure

In this section, two neighborhood structures are presented for the FJSP. The first neighborhood focuses mainly on the sequencing of operations and random assignments of machines to each operation, and the second neighborhood focuses on a local search neighborhood to improve the assignment of the machines once the sequence of operations has been modified.

For the second neighborhood, a simplification of the Nopt1 neighborhood and the makespan estimation explained in *Mastrolilli & Gambardella (2000)* are used to improve the GLNSA execution time.

## Exploration neighborhood

For the global search neighborhood, well-known operators used in various task sequencing problems are used. These operators are used to optimize the sequence of operations. For the machine assignment, the mutation operator used in *Li & Gao (2016)* is applied. Each smart-cell generates $l$ neighbors using one of the three possible operators (insertion, swapping, or PR) with probability $\alpha_I$, $\alpha_S$ and $\alpha_P$, respectively, to generate a variant of $OS$. For the machine assignment, the mutation operator with probability $\alpha_M$ is employed to generate another variant of $MS$. From these $l$ neighbors, the one with the smallest makespan is chosen to be the new smart-cell. This neighborhood is exemplified in Fig. 3.

### Insertion

The insertion operator consists of selecting two different positions, $k_1$ and $k_2$, of the string $OS$ in a smart-cell to obtain another string $OS'$. For example, if $k_1 > k_2$ with $OS = (O_1, \ldots Ok_2 \ldots O k_1 \ldots)$, then we get the string $OS' = (O_1, \ldots Ok_1 Ok_2 \ldots Ok_1 - 1 \ldots)$. This is analogous if $k_1 < k_2$. This is exemplified in Fig. 4.

### Swapping

The swapping operator consists of selecting 2 different positions, $k_1$ and $k_2$, from the string $OS$ to exchange their positions. For example, if $k_1 < k_2$ with $OS = (O_1, \ldots Ok_1 \ldots Ok_2 \ldots)$, then the string $OS' = (O_1, \ldots Ok_2 \ldots Ok_1 \ldots)$. This is analogous if $k_1 > k_2$. This is exemplified in Fig. 5.

Insertion and swapping are operators classically used in metaheuristics for task scheduling problems, since they provide good quality solutions (*Błażewicz, Domschke & Pesch, 1996*; *Cheng, Gen & Tsujimura, 1999*; *Deroussi, Gourgand & Norre, 2006*).

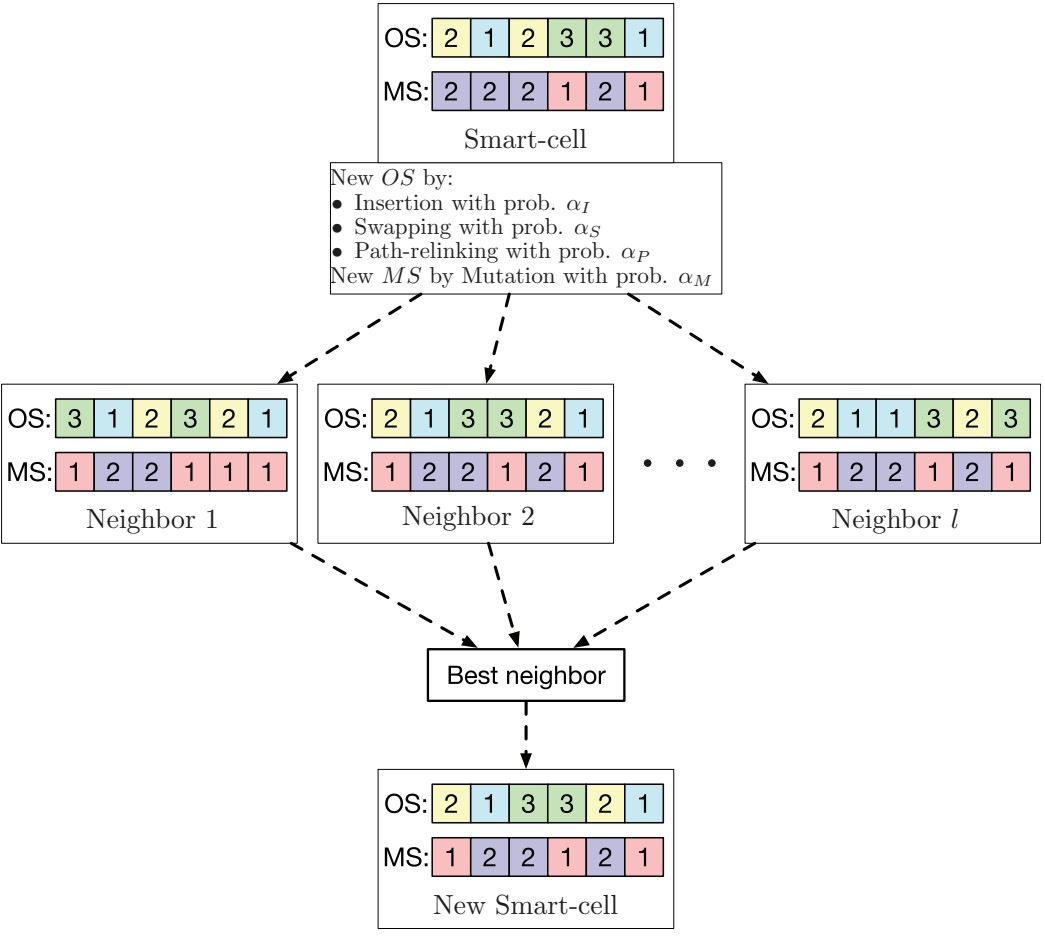

**Figure 3 Exploration neighborhood used by the GLNSA.**

## Path relinking

Path relinking (PR) involves establishing a route between two different smart-cells for their strings $OS$ and $OS'$. The route defines intermediate strings ranging from $OS$ to $OS'$. To form this route, the first position $k$ of $OS$ is taken such that $OS_k \neq OS'_k$. Then, the first position $p$ is located after $k$ such that $OS_p = OS'_k$. Once both positions have been found, the positions $k$ and $p$ are exchanged in $OS$ to obtain a new string closer to $OS'$. The PR is repeated until $OS'$ is obtained. In the end, one of the intermediate solutions is randomly chosen.

The idea of the PR is to generate solutions that combine the information of $OS$ and $OS'$ and fulfill two objectives in the GLNSA: if both smart-cells have similar machine strings, PR acts as a local search method that refines the strings of operations. Conversely, if both smart-cells have very different machine strings, PR works as an exploration method that generates new variants of one of the smart-cells, taking the other as a guide, as shown in Fig. 6. PR has already been used successfully in the FJSP, as shown in its application with different neighborhood variants in *González, Vela & Varela (2015)*.

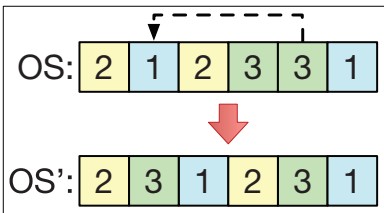

**Figure 4 The insertion operator.**

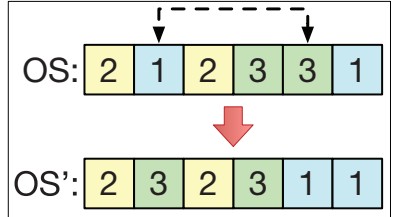

**Figure 5 The swapping operator.**

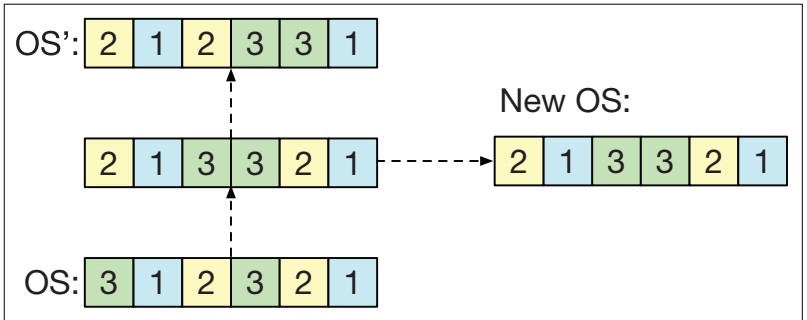

**Figure 6 The path relinking operator.**

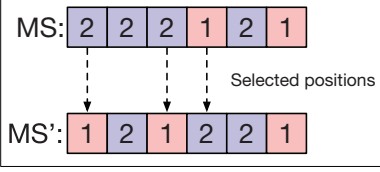

**Figure 7 Mutation operator in machine strings.**

## Mutation of the string of machines

For the *MS* string of each smart-cell, a mutation operator is applied that selects half of the positions in *MS* at random. For each position linked to an operation in *OS*, the assigned machine is changed for another selected at random so that it can perform the respective operation in *OS*.

Figure 7 shows an example of the mutation operator applied to a sequence *MS*. Three machines are selected corresponding to the operations $O_{11}$, $O_{21}$, and $O_{22}$ in *OS*. The new

machine assignment is obtained by changing the machines in these positions to others that can also perform these operations.

## Local search neighborhood

Simple local search methods are based on creating slight modifications of a solution to generate an improved one until the process can no longer achieve an improvement, or a certain number of repetitions is met. These methods tend to get trapped in local minima or regions where many solutions have the same cost. In order to make an algorithm more adaptable and capable of escaping local optima, various strategies have been proposed. One of the most effective ones so far is the TS.

TS is a metaheuristic developed by *Glover & Laguna (1998)* to guide a local search more effectively by incorporating an adaptive memory and a more intelligent exploration. It is one of the most used methods and offers satisfactory results in many optimization problems (*Glover & Kochenberger, 2006*). TS allows, during the search process, solutions with a worse cost to be accepted to provide a more significant search capacity. A short-term memory prohibits returning to solutions that have already been recently explored. They are again taken if they are outside the tabu threshold or meet an aspiration criterion (*Chaudhry & Khan, 2016*). In this way, the search does not have a cyclical behavior and produces new trajectories in the solution space.

In the GLNSA, TS allows generating solutions that may not improve the makespan of the original smart-cell as long as the operation and the machine selected to obtain the new solution are not forbidden. TS keeps a record of the operation and the machine selected in each movement and the threshold at which this movement will remain tabu, and the aspiration criterion allows a solution to be accepted even if it is tabu.

The implementation of TS is resumed using a simplification of the neighborhood structure proposed in *Mastrolilli & Gambardella (2000)* and the makespan estimation explained in the same work to reduce the computational time of TS. The general TS procedure is described in Algorithm 2.

The flow chart of the TS is shown in Fig. 8.

## Critical path

The TS for this job uses the following definition of the critical path *cp*. To form *cp*, one of the operations with completion time equal to the makespan is selected randomly. Once this operation has been chosen, the operation that precedes it is selected either on the same machine or by the previous operation of the same job, and the one whose completion time is equal to the initial time of the current operation is selected. If both previous operations have the same completion time, one of them is selected randomly. This process is repeated until an operation with a start time of 0 is selected. These operations form a critical path *cp* of length *q*.

## Simplified Nopt1 neighborhood

The neighborhood used for the local search is based on the one defined as Nopt1 in *Mastrolilli & Gambardella (2000)*. Given a critical path *cp*, in the original neighborhood

---

**Algorithm 2** General description of the TS.

**Result**: Best smart-cell

Take a smart-cell as initial solution;

Take the operations and their machine assignments from a critical path of the smart-cell;

Initialize TS (empty list of operations/machines, iteration number $T_n$ for the TS, and tabu threshold $T_u$);

Set *new_solution = smart_cell*;

**for** $T_n$ iterations **do**

> Generate new neighborhoods of *new_solution* modifying only the assigned machines to the critical operations;
>
> Select the *best_neighbor*;
>
> **if** *best_neighbor holds the aspiration criterion or is not tabu* **then**
>
> > Set *best_neighbor* as *new_solution*;
>
> **else**
>
> > Select the oldest tabu neighbor as *new_solution*;
>
> **end**
>
> Continue the TS from this *new_solution*;
>
> Update the TS list by setting the selected entry operation/machine with the sum of the threshold $T_u$ plus the current iteration, increasing the number of iterations;

**end**

Return the best *new_solution* as new smart-cell;

---

Nopt1, for each operation in *cp*, a set of preceding and succeeding operations can be found in each feasible machine, such that a new placement of the operation between these operations on the new machine optimizes the makespan. The calculation of Nopt1 depends on the review of the start and tail times of the operations programmed in each machine, implying an almost constant computational time. When this operation is performed several times (such as in a meta-heuristic algorithm), the computational time can increase considerably, especially if the number of jobs and machines is large and the system has high flexibility.

In this work, it is proposed to use a simplification of the Nopt1 neighborhood, where a search for the best position is not carried out to accommodate each operation in a new feasible machine, but its machine assignment is just changed in the string *MS*, preserving the position in the string *OS*. The idea is that the global search operations applied in the neighborhood based on cellular automata (insertion, swapping, and PR) are capable of finding this optimal assignment as the optimization algorithm advances, especially for systems with greater flexibility, where more machines are capable of processing the same operation. Thus, the objective is to avoid carrying out the operations that explicitly seek to accommodate an operation on a machine and directly take the position that is being refined by the global neighborhood's operations.

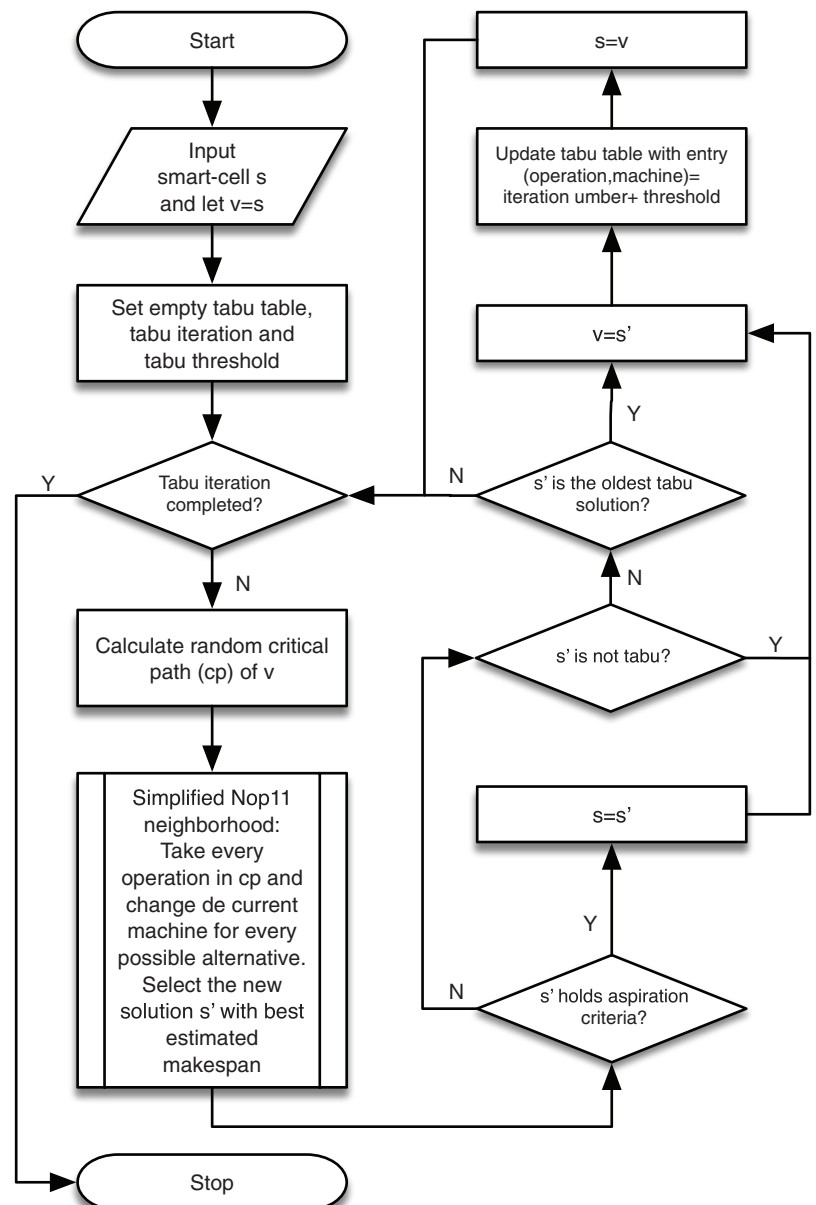

**Figure 8 TS applied to the local search process of the GLNSA.**

There are more optimal job placement positions for systems with high flexibility, given the greater availability of feasible machines, so this simplification focuses on showing that good results can be obtained with a reduced process for this type of FJSP instance.

## Parameters of the GLNSA

The following are the parameters of the proposed algorithm:

- Number of iterations for the whole optimization process: $G_n$.
- Number of smart-cells: $S_n$.

- Global neighborhood size: $l$.
- Probabilities $\alpha_I$, $\alpha_S$, and $\alpha_P$ for insertion, swapping, and PR, respectively, and probability $\alpha_M$ for mutation.
- Maximum number of stagnation iterations: $S_b$.
- Proportion of elitist solutions: $E_p$.
- Number of tabu iterations for every optimization iteration: $T_n$.
- Tabu threshold: $T_u$.

## Parameter tuning

A preliminary study was carried out considering different values of the GLNSA parameters. They were applied to the same problem using a similar number of iterations to select the best parameters applied to instances of FJSP with high flexibility.

For the population size $S_n$, the number of iterations $G_n$, and the mutation probability $\alpha_M$, the results presented in *González, Vela & Varela (2015)* and *Li & Gao (2016)* are taken as a basis, since they are recent works that show great effectiveness both in the makespan calculation as well as in the runtime for FJSP instances.

For $G_n$, values of 200 and 250 were tested. Also, while $S_n$ is taken between 20 and 100 in *González, Vela & Varela (2015)* and defined as 400 in *Li & Gao (2016)*, we tested $S_n$ between 40 and 80.

The number of neighbors $l$ that each smart-cell has in our algorithm to generate the global search neighborhood was tested with values between 2 and 3, in order to preserve a population close to 250 solutions at most (number of smart-cells by number of neighbors) and keep a computational execution close to the cited references. To form the global neighborhood, the probability combinations ($\alpha_I$, $\alpha_S$, $\alpha_P$) with respective values (0.5,0.25,0.25), (0.25,0.5,0.25) and (0.25,0.25, 0.5) were tested. The mutation probability $\alpha_M$ was taken with values 0.1 and 0.2.

To control the stagnation limit $S_b$, the value proposed by *Li & Gao (2016)* is taken to test $S_b$ with values 20 and 40, and for the elitist proportion of solutions, a value of $E_p$ of 0.025 and 0.05 is considered.

Without a doubt, the TS is the most computationally expensive process that the proposed algorithm has. In *González, Vela & Varela (2015)*, the execution of the TS is tested up to 10,000 iterations per solution. In *Li & Gao (2016)*, this number goes up to about 80,000 times per solution, of course, with different ways of creating solutions and estimating the makespan. Our algorithm uses the makespan estimation developed in *Mastrolilli & Gambardella (2000)* during the TS to reduce the computational time of the optimization process.

In our algorithm, we take a point of view similar to *Li & Gao (2016)*, using an increasing number of iterations of TS, as the number of iterations of the optimization process grows. For each iteration $i$, $T_n * i$ TS iterations will be taken for each smart-cell, where $T_n$ values of 1 and 2 are tested to keep a maximum TS close to 60,000 iterations per smart-cell. Altogether, it took 384 different combinations of parameters to tune the GLNSA.

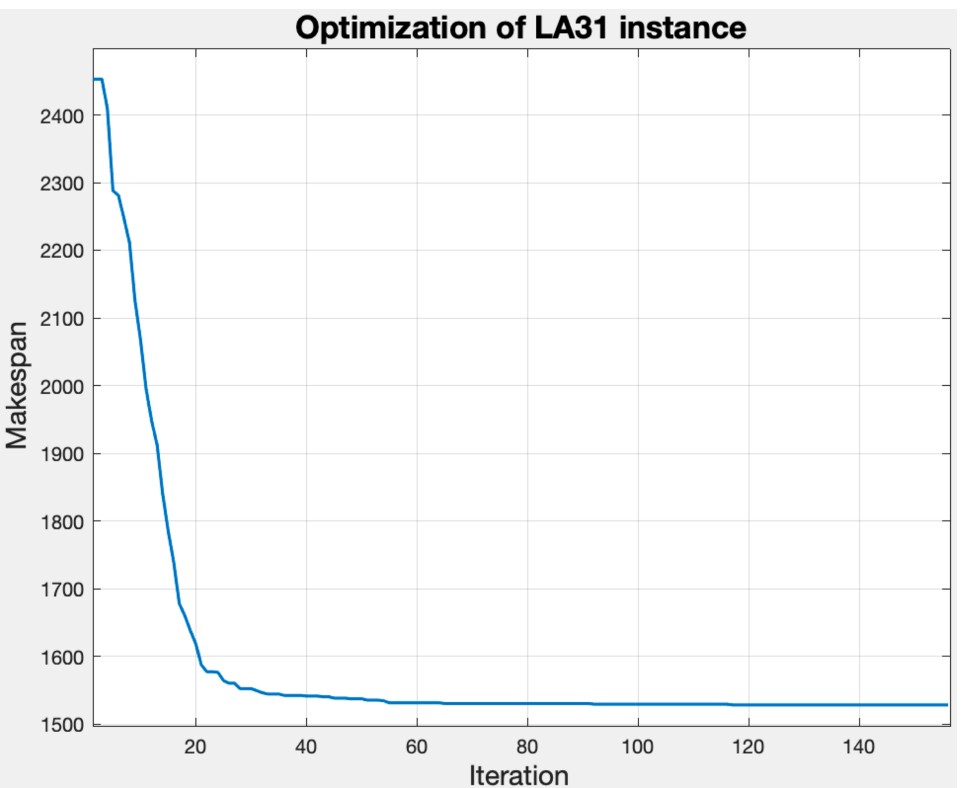

**Figure 9 Behavior of the makespan with tuned parameters for the la31-vdata instance.**

For this preliminary tuning study, we took the problem *la*31 with more flexibility (known as part of the vdata set), initially proposed by *Hurink, Jurisch & Thole (1994)*, and downloaded from https:/people.idsia.ch/monaldo/fjsp.html. From the tuning study, the parameters $G_n = 250$, $S_n = 40$, $l = 2$, $\alpha_I = 0.5$, $\alpha_S = 0.25$, $\alpha_P = 0.25$, $\alpha_M = 0.1$, $S_b = 40$; $E_p = 0.025$ and $T_n = 1$ were selected as the most appropriate values to apply the GLNSA.

The threshold used in the tabu list for each entry (operation/machine) consists of the sum of the length of the random critical path plus the number of feasible machines that can perform the corresponding operation. This criterion was also proposed in *Mastrolilli & Gambardella (2000)* and has been widely used by similar algorithms.

Figure 9 shows the convergence of the makespan by applying the GLNSA to the parameters indicated above on the instance *la*31-vdata. The implementation was developed in Matlab (the implementation characteristics and computational experiments are specified in detail in the next section), and the execution time was 56 s.

## EXPERIMENTAL RESULTS

The GLNSA was implemented in Matlab R2015a (TM) on a 2.3 GHz Intel Xeon W machine and with 128 GB of RAM. The GLNSA was compared with other nine algorithms, proposed between 2015 and 2020, to show the effectiveness of the proposed method. These algorithms are the efficient PSO and gravitational search algorithm (ePSOGSA) (*Bharti & Jain, 2020*), the greedy randomized adaptive search procedure (GRASP)

(*Baykasoğlu, Madenoğlu & Hamzaday, 2020*), the HA (*Li & Gao, 2016*), the improved Jaya algorithm (IJA) (*Caldeira & Gnanavelbabu, 2019*), the self-learning GA (SLGA) (*Chen et al., 2020*), the scatter search with PR algorithm (SSPR) (*González, Vela & Varela, 2015*), the teaching-learning-based optimization (TLBO) (*Buddala & Mahapatra, 2019*), the two-level PSO (TlPSO) (*Zarrouk, Bennour & Jemai, 2019*), and the VNS-based GA (VNSGA) (*Zhang et al., 2019*).

To test the efficiency of the GLNSA, 4 benchmark datasets commonly used in the FJSP literature were taken, each with a different number of instances. The flexibility of each instance was weighted with a rate $\beta$ = (flexibility average/number of machines). The rate $\beta$ is between 0 and 1. A higher value indicates that more machines can perform more different operations. The first dataset is the Kacem dataset (*Kacem, Hammadi & Borne, 2002a*; *Kacem, Hammadi & Borne, 2002b*), with 5 instances of different characteristics and 4 with $\beta$ = 1. In other words, all machines can perform all operations, showing a system with total flexibility. The second benchmark dataset is the Brandimarte dataset (*Brandimarte, 1993*), with 10 instances of partial flexibility ($\beta < 1$). The last dataset is the Hurink dataset (*Hurink, Jurisch & Thole, 1994*), from which two groups of 43 problems each are taken, the first with low partial flexibility and the second with high partial flexibility. These four datasets make a total of 101 different instances to test and compare the efficiency of the GLNSA. For the Kacem and Brandimarte benchmarks, almost all the referenced algorithms present results, but only a few for all the instances. For the Hurink benchmark, only four algorithms present results (ePSOGSA, HA, IJA, and SSPR), against which the GLNSA is compared for this case.

## Comparative analysis of computational complexity between algorithms

Since it is difficult to compare the execution time of each of these algorithms since they were tested in different architectures, languages, and programming skills, these algorithms are compared based on their computational complexity $\mathcal{O}(o)$, taking as a reference the total number of operations $o = \sum_{i=1}^{n} \sum_{j=1}^{n_i} O_{i,j}$ to be processed for an instance of the FJSP. To generalize the way of measuring each algorithm's complexity, we will use the notation defined for the GLNSA parameters; the number of iterations will be denoted by $G_n$ and the number of solutions handled by each algorithm will be defined by $X$, which in the case of the GLNSA is $X = S_n l$. For the local search each algorithm applies, the number of iterations is expressed with $T_n$, and for the number of machines, $m$ will be used.

The ePSOGSA first uses a modified PSO to update each individual by rate or mutation, then applies a modified GSA to obtain an extended population and sort a final population. The ePSOGSA does not use an iterative local search based on the calculation of critical paths. The GRASP algorithm constructs an optimized Gantt chart and then applies a greedy local search that is quadratic concerning the number of operations. The HA has an architecture similar to the GLNSA proposed in this work, and first applies four types of genetic operators to each solution. Then, HA applies a TS that is supported by calculating the critical path and selecting different machines for each operation. The IJA uses a modified Java algorithm that applies three change operators to each solution and then, in

**Table 1 Algorithms used in the experiments and their computational complexity.**

| Algorithm | Complexity | Rank |
|---|---|---|
| SLGA | $\mathcal{O}(o(G_n 5X))$ | 1 |
| ePSOGA | $\mathcal{O}(o(G_n 8X) + X log X)$ | 2 |
| TlPSO | $\mathcal{O}(o(G_n 2X) + G_n 2X)))$ | 3 |
| SSPR | $\mathcal{O}(o(X T_n m + Gn(X + T_n m)))$ | 4 |
| GLNSA | $\mathcal{O}(o(G_n(S_n + X + S_n T_n m)))$ | 5 |
| IJA | $\mathcal{O}(o(G_n(3X + X T_n m))$ | 6 |
| HA | $\mathcal{O}(o(G_n(4X + X T_n m)))$ | 7 |
| TLBO | $\mathcal{O}(o(G_n(4X + X T_n m)$ | 7 |
| VNSGA | $\mathcal{O}(o(G_n(3X + 4X T_n m)))$ | 8 |
| GRASP | $\mathcal{O}(o^2(G_n * T_n))$ | 9 |

the same cycle, a local search to all individuals based on the critical path and random exchange of critical blocks in the machines, obtaining a complexity very similar to HA. The SLGA uses two genetic operators for each individual and then uses greedy reinforcement learning based on selecting suitable individuals and two types of actions that also do not need to calculate a critical path. The SSPR algorithm also has a similar structure to HA and GLNSA. It first uses a TS for each individual and then uses PR between two pairs of solutions iteratively to obtain a new individual, which is again improved by a TS based on the critical path calculation. The SSPR also employs a diversification phase when all pairs of individuals have been selected for the PR. The TLBO algorithm is based on a teaching-learning algorithm that uses real coding that utilizes three learning rules and must review the feasibility of the solutions. It also applies a local search based on the exchange of critical operations and their change of machines. The TlPSO algorithm applies two modifications to the PSO in order to optimize the routing problem in the first stage, and within this stage, optimizes the machine assignment problem with another PSO. The last algorithm taken for comparison was the VNSGA, which first applies three genetic operators to each individual and then performs a local search based on the critical path of each solution where processing-time conditions are detected for four neighborhood types available for the machines assigned to each solution.

Finally, the algorithm proposed in this work (GLNSA) performs an elitist selection of solutions, then a search for $l$ based on simple operators such as insertion, swapping, PR, and machine mutation on the set of $S_n$ smart-cells and then a TS only on $S_n$ smart-cells. Since $S_n < X$, the GLNSA performs less computation for the local search.

Table 1 presents the algorithms used for comparison in this work ordered depending on their complexity, taking as a reference that $S_n < X < T_n$, since in the algorithms that apply a local search, a high value of $T_n$ is usually chosen for best results. We can see that the GLNSA has a competitive computational complexity compared to state-of-art algorithms recently proposed in the literature for the FJSP.

It should be noted that this analysis only considers the computational complexity required to manipulate and modify the scheduling and routing of the sequences of operations of an FJSP instance. The computational complexity for the makespan

**Table 2 Experimental results for the Kacem instances.**

| Instance | n × m | β | ePSOGA | GLNSA | GRASP | HA | IJA | SLGA | TLBO | TlPSO | VNSGA |
|----------|-------|------|--------|-------|-------|----|-----|------|------|-------|-------|
| K1 | 4 × 5 | 1 | 11 | 11 | – | – | 11 | 11 | 11 | 11 | – |
| K2 | 8 × 8 | 0.81 | – | 14 | 14 | 14 | 14 | 14 | 14 | 14 | 14 |
| K3 | 10 × 7 | 1 | – | 11 | – | – | 11 | 11 | 11 | – | – |
| K4 | 10 × 10 | 1 | 7 | 7 | 7 | 7 | 7 | – | 7 | 7 | 7 |
| K5 | 15 × 10 | 1 | 11 | 11 | 11 | 11 | 11 | – | 13 | – | 11 |

calculation is $\mathcal{O}(o^2)$ and the makespan estimation taking into account only critical operations is bounded by $\mathcal{O}(o)$. These processes are not contemplated in the analysis presented since all the algorithms use these operations, and we only focus on the study of the computational process that distinguishes each algorithm.

## First experiment, Kacem instances

Table 2 presents the GLNSA results and the comparison with algorithms that present results for the Kacem benchmark. $n$ represents the number of jobs and $m$ the number of machines, in addition to the flexibility rate $\beta$ of each instance. This benchmark dataset is characterized by its high flexibility and starts from instances with low dimensionality to instances with a greater number of jobs and machines.

For this benchmark, only the IJA and TLBO algorithms report complete results (*Caldeira & Gnanavelbabu, 2019*; *Buddala & Mahapatra, 2019*). In this experiment, the GLNSA obtains the best-known results for the makespan value in all cases, like the IJA, and improves the TLBO for the K5 instance with greater dimensionality. This experiment corroborates the excellent performance of the GLNSA for problems with a high rate of flexibility $\beta$.

## Second experiment, Brandimarte instances

To compare the results of the nine algorithms in this benchmark, the relative percentage deviation (*RPD*) is defined in Eq. 7.

$$RPD = \frac{BOV - BKV}{BOV} \times 100 \tag{7}$$

where BOV is the best value obtained by the algorithm, and BKV is the best-known value for each instance, in the Brandimarte benchmark, and the best-reported values are taken from *Chen et al. (2020)*. Table 3 presents the results of the GLNSA and its comparison with the other algorithms. In this table, the rate $\beta$ of each instance is presented, which varies from 0.15 to 0.35. All cases have partial flexibility. The best makespan obtained by any of the algorithms is indicated with a *. In this experiment, it can be observed that the GLNSA obtains the best makespan 6 times, only behind the ePSOGA and the SSPR, and shares the number of best results obtained with the GRASP, HA, and TlPSO algorithms.

Table 4 presents the average *RPD* of each algorithm in the 10 instances and the ranking of each algorithm, taking the average *RPD* as a reference. The GLNSA is seen to obtain the

**Table 3 Experimental results for the Brandimarte instances.**

| Instance | n × m | β | BKV | ePSOGA | GLNSA | GRASP | HA | IJA | SLGA | SSPR | TLBO | TlPSO | VNSGA |
|---|---|---|---|---|---|---|---|---|---|---|---|---|---|
| MK01 | 10 × 6 | 0.2 | 36 | 40 | 40 | 40 | 40 | 40 | 40 | 40 | 39* | 40 | 40 |
| MK02 | 10 × 6 | 0.35 | 24 | 26* | 26* | 26* | 26* | 27 | 27 | 26* | 27 | 26* | 27 |
| MK03 | 15 × 8 | 0.3 | 204 | 204* | 204* | 204* | 204* | 204* | 204* | 204* | 204* | 204* | 204* |
| MK04 | 15 × 8 | 0.2 | 48 | 60* | 60* | 60* | 60* | 60* | 60* | 60* | 63 | 60* | 60* |
| MK05 | 15 × 4 | 0.15 | 168 | 170* | 173 | 172 | 172 | 172 | 172 | 172 | 172 | 173 | 173 |
| MK06 | 10 × 15 | 0.3 | 33 | 56* | 58 | 64 | 57 | 57 | 69 | 57 | 65 | 60 | 58 |
| MK07 | 20 × 5 | 0.3 | 133 | 139* | 139* | 139* | 139* | 139* | 144 | 139* | 144 | 139* | 144 |
| MK08 | 20 × 10 | 0.15 | 523 | 523* | 523* | 523* | 523* | 523* | 523* | 523* | 523* | 523* | 523* |
| MK09 | 20 × 10 | 0.3 | 299 | 307* | 307* | 307* | 307* | 307* | 320 | 307* | 311 | 307* | 307* |
| MK10 | 20 × 15 | 0.2 | 165 | 196* | 205 | 205 | 197 | 197 | 254 | 196* | 214 | 205 | 198 |

**Note:**
* Best obtained makespan.

**Table 4 Average RPD and Friedman test for Brandimarte instances.**

| Algorithm: | ePSOGA | GLNSA | GRASP | HA | IJA | SLGA | SSPR | TLBO | TlPSO | VNSGA | p value |
|---|---|---|---|---|---|---|---|---|---|---|---|
| Average RPD: | 10.2679 | 11.0121 | 11.4890 | 10.5289 | 10.8708 | 14.4851 | 10.4862 | 13.4046 | 11.3128 | 11.0597 | 0.0003 |
| Rank: | 1 | 5 | 8 | 3 | 4 | 10 | 2 | 9 | 7 | 6 | |

fifth position among the algorithms taken for comparison, which shows its competitiveness with state-of-the-art algorithms for this benchmark.

A non-parametric Friedman test was performed with the *RPD* values in all instances to corroborate whether there is a statistically significant comparison between the results obtained by all the algorithms (*Derrac et al., 2011*). The result of the Friedman test is observed in the last column of Table 4 with a value of $p = 0.0003$, which is less than the significant level of 0.05 to reject the null hypothesis that algorithms behave statistically similarly. The $p$ value demonstrates significant differences in the performance of the 10 algorithms, showing the GLNSA's competitiveness to optimize this benchmark.

### Third experiment, Hurink instances with low flexibility

This experiment takes 43 instances from the Hurink benchmark (rdata) with low flexibility, whose rate $\beta \leq 0.4$. The results of the GLNSA are compared taking the BKVs of every instance as reported in *Li & Gao (2016)*. Table 5 shows the comparison of the GLNSA with the previous algorithms that report results for this benchmark (ePSOGA, HA, IJA, and SSPR); the ePSOGA only reports 15 results from the 43 instances, so only these results were taken to complete the comparative analysis. The results marked with * are the best obtained among the five algorithms.

Table 6 presents the average *RPD* of the algorithms that report complete results for the 43 instances and the ranking of each algorithm, taking the average *RPD* as a reference. The GLNSA is seen to obtain the fourth position among the algorithms taken for comparison, and the Friedman test obtains a value of $p = 0.000000002$, which is less than

**Table 5 Experimental results for the Hurink-rdata instances.**

| Instance | n × m | β | BKV | ePSOGSA | GLNSA | HA | IJA | SSPR |
|---|---|---|---|---|---|---|---|---|
| mt06 | 6 × 6 | 0.33 | 47 | – | 47* | 47* | 47* | 47* |
| mt10 | 10 × 10 | 0.2 | 686 | – | 686 * | 686 * | 686* | 686* |
| mt20 | 20 × 5 | 0.4 | 1,022 | – | 1,022 * | 1,024 | 1,024 | 1,022* |
| la01 | 10 × 5 | 0.4 | 570 | 572 | 571 | 570 * | 571 | 571 |
| la02 | 10 × 5 | 0.4 | 529 | 529* | 530 | 530 | 530 | 530 |
| la03 | 10 × 5 | 0.4 | 477 | 478 | 477* | 477* | 477* | 477* |
| la04 | 10 × 5 | 0.4 | 502 | 502* | 502* | 502* | 502* | 502* |
| la05 | 10 × 5 | 0.4 | 457 | 457* | 457* | 457* | 457* | 457* |
| la06 | 15 × 5 | 0.4 | 799 | 800 | 799* | 799* | 799* | 799* |
| la07 | 15 × 5 | 0.4 | 749 | 750 | 749* | 749* | 749* | 749* |
| la08 | 15 × 5 | 0.4 | 765 | 765* | 765* | 765* | 765* | 765* |
| la09 | 15 × 5 | 0.4 | 853 | 853* | 853* | 853* | 853* | 853* |
| la10 | 15 × 5 | 0.4 | 804 | 805 | 804* | 804* | 804* | 804* |
| la11 | 20 × 5 | 0.4 | 1,071 | 1,071* | 1,071* | 1,071* | 1,071* | 1,071* |
| la12 | 20 × 5 | 0.4 | 936 | 936* | 936 * | 936* | 936* | 936* |
| la13 | 20 × 5 | 0.4 | 1,038 | 1,038* | 1,038* | 1,038* | 1,038* | 1,038* |
| la14 | 20 × 5 | 0.4 | 1,070 | 1,070* | 1,070* | 1,070* | 1,070* | 1,070* |
| la15 | 20 × 5 | 0.4 | 1,089 | 1,090 | 1,089* | 1,090 | 1,090 | 1,089* |
| la16 | 10 × 10 | 0.2 | 717 | – | 717* | 717* | 717* | 717* |
| la17 | 10 × 10 | 0.2 | 646 | – | 646* | 646* | 646* | 646* |
| la18 | 10 × 10 | 0.2 | 666 | – | 666* | 666* | 666* | 666* |
| la19 | 10 × 10 | 0.2 | 647 | – | 700* | 700* | 702 | 700* |
| la20 | 10 × 10 | 0.2 | 756 | – | 756* | 756* | 760 | 756* |
| la21 | 15 × 10 | 0.2 | 808 | – | 852 | 835 | 854 | 830* |
| la22 | 15 × 10 | 0.2 | 737 | – | 774 | 760 | 760 | 756* |
| la23 | 15 × 10 | 0.2 | 816 | – | 854 | 840 | 852 | 835* |
| la24 | 15 × 10 | 0.2 | 775 | – | 826 | 806 | 806 | 802* |
| la25 | 15 × 10 | 0.2 | 752 | – | 803 | 789 | 803 | 784* |
| la26 | 20 × 10 | 0.2 | 1,056 | – | 1,075 | 1,061 | 1,061 | 1,059* |
| la27 | 20 × 10 | 0.2 | 1,085 | – | 1,109 | 1,089* | 1,109 | 1,089* |
| la28 | 20 × 10 | 0.2 | 1,075 | – | 1,096 | 1,079 | 1,081 | 1,078* |
| la29 | 20 × 10 | 0.2 | 993 | – | 1,008 | 997 | 997 | 996* |
| la30 | 20 × 10 | 0.2 | 1068 | – | 1,096 | 1,078 | 1,078 | 1,074* |
| la31 | 30 × 10 | 0.2 | 1520 | – | 1,527 | 1,521 | 1,521 | 1,520* |
| la32 | 30 × 10 | 0.2 | 1,657 | – | 1,667 | 1,659 | 1,659 | 1,658 |
| la33 | 30 × 10 | 0.2 | 1,497 | – | 1,504 | 1,499 | 1,499 | 1,498* |
| la34 | 30 × 10 | 0.2 | 1,535 | – | 1,540 | 1,536 | 1,536 | 1,535* |
| la35 | 30 × 10 | 0.2 | 1,549 | – | 1,555 | 1,550* | 1,555 | 1,550* |
| la36 | 15 × 15 | 0.13 | 1,016 | – | 1,053 | 1,028 | 1,050 | 1,023* |
| la37 | 15 × 15 | 0.13 | 989 | – | 1,093 | 1,074 | 1,092 | 1,069* |
| la38 | 15 × 15 | 0.13 | 943 | – | 999 | 960* | 995 | 961 |
| la39 | 15 × 15 | 0.13 | 966 | – | 1,034 | 1,024* | 1,031 | 1,024* |
| la40 | 15 × 15 | 0.13 | 955 | – | 997 | 970 | 993 | 961* |

**Note:**
  * Best obtained makespan.

**Table 6 Average RPD and Friedman test for all the Hurink-rdata instances.**

| Algorithm: | GLNSA | HA | IJA | SSPR | p value |
|---|---|---|---|---|---|
| Average *RPD*: | 1.7768 | 1.0872 | 1.5155 | 0.9566 | 0.000000002 |
| Rank: | 4 | 2 | 3 | 1 | |

**Table 7 Average RPD and Friedman test for the Hurink-rdata instances with greater flexibility.**

| Algorithm: | ePSOGSA | GLNSA | HA | IJA | SSPR | p value |
|---|---|---|---|---|---|---|
| Average *RPD*: | 0.0689 | 0.0243 | 0.0187 | 0.0304 | 0.0243 | 0.0339 |
| Rank: | 5 | 2 | 1 | 4 | 3 | |

the significant level of 0.05, proving that there are significant differences in the performance of the 4 algorithms. The low competitiveness of the GLNSA with respect to the other algorithms can be explained by the results in instances la21 to la40, which correspond to the examples with higher dimensionality (number of jobs and machines) and lower flexibility. This is expected given the functioning of the GLNSA that is focused on resolving FJSP instances with a higher rate $\beta$.

However, if we analyze only the problems with greater flexibility (la01 to la15) to consider the ePSOGSA algorithm and do the same average *RPD* examination and Friedman test, we obtain the result shown in Table 7. For these instances with more flexibility, the GLNSA ranks second among the 5 algorithms with a value of $p = 0.0339$, which is less than the significant level of 0.05, demonstrating significant differences in performance of the 5 algorithms for these instances. The preceding results corroborate the efficiency of the GLNSA to optimize FJSP instances with high flexibility.

## Fourth experiment, Hurink instances with greater flexibility

The fourth experiment takes 43 instances of the Hurink-vdata benchmark with rate $\beta = 0.5$, indicating that an operation can be processed by around half the machines, already implying a high degree of flexibility. The results of the GLNSA are compared with the algorithms taken before that report results for this benchmark (HA, IJA, and SSPR) in Table 8, taking the BKVs for statistical analysis again from *Li & Gao (2016)*.

Table 9 presents the average *RPD* of the algorithms and their ranking. According to this average, the GLNSA obtains the third position among the algorithms taken for comparison. Two non-parametric Friedman tests were made, one among the 4 algorithms and the other only comparing the GLNSA with the HA, which are the ones that had the closest values.

In Table 9, a value of $p = 0.0000003$ is obtained, which is less than the significant level of 0.05, proving that there are significant differences in the performance of the 4 algorithms. However, when comparing only the GLNSA to the HA, a value of $p = 0.5637$ is obtained, which rejects a significant difference between both algorithms to optimize this benchmark dataset. This test verifies that the efficiency of the GLNSA is similar to the HA and is only significantly below the SSPR. These results confirm the observation that the simplified

**Table 8 Experimental results for the Hurink-vdata instances.**

| Instance | n × m | β | BKV | GLNSA | HA | IJA | SSPR |
|---|---|---|---|---|---|---|---|
| mt06 | 6 × 6 | 0.5 | 47 | 47* | 47* | 47* | 47* |
| mt10 | 10 × 10 | 0.5 | 655 | 655* | 655* | 655* | 655* |
| mt20 | 20 × 5 | 0.5 | 1022 | 1022* | 1022* | 1024 | 1022* |
| la01 | 10 × 5 | 0.5 | 570 | 570* | 570* | 571 | 570* |
| la02 | 10 × 5 | 0.5 | 529 | 529* | 529* | 529* | 529* |
| la03 | 10 × 5 | 0.5 | 477 | 477* | 477* | 477* | 477* |
| la04 | 10 × 5 | 0.5 | 502 | 502* | 502* | 502* | 502* |
| la05 | 10 × 5 | 0.5 | 457 | 457* | 457* | 457* | 457* |
| la06 | 15 × 5 | 0.5 | 799 | 799* | 799* | 799* | 799* |
| la07 | 15 × 5 | 0.5 | 749 | 749* | 749* | 749* | 749* |
| la08 | 15 × 5 | 0.5 | 765 | 765* | 765* | 765* | 765* |
| la09 | 15 × 5 | 0.5 | 853 | 853* | 853* | 853* | 853* |
| la10 | 15 × 5 | 0.5 | 804 | 804* | 804* | 804* | 804* |
| la11 | 20 × 5 | 0.5 | 1071 | 1071* | 1071* | 1071* | 1071* |
| la12 | 20 × 5 | 0.5 | 936 | 936* | 936* | 936* | 936* |
| la13 | 20 × 5 | 0.5 | 1038 | 1038* | 1038* | 1038* | 1038* |
| la14 | 20 × 5 | 0.5 | 1070 | 1070* | 1070* | 1070* | 1070* |
| la15 | 20 × 5 | 0.5 | 1089 | 1089* | 1089* | 1089* | 1089* |
| la16 | 10 × 10 | 0.5 | 717 | 717* | 717* | 717* | 717* |
| la17 | 10 × 10 | 0.5 | 646 | 646* | 646* | 646* | 646* |
| la18 | 10 × 10 | 0.5 | 663 | 663* | 663* | 665 | 663* |
| la19 | 10 × 10 | 0.5 | 617 | 617* | 617* | 618 | 617* |
| la20 | 10 × 10 | 0.5 | 756 | 756* | 756* | 758 | 756* |
| la21 | 15 × 10 | 0.5 | 800 | 806 | 804* | 806 | 804* |
| la22 | 15 × 10 | 0.5 | 733 | 737* | 738 | 738 | 738 |
| la23 | 15 × 10 | 0.5 | 809 | 813 | 813 | 813 | 812* |
| la24 | 15 × 10 | 0.5 | 773 | 777 | 777 | 778 | 775* |
| la25 | 15 × 10 | 0.5 | 751 | 754* | 754* | 754* | 754* |
| la26 | 20 × 10 | 0.5 | 1052 | 1054 | 1053* | 1054 | 1053* |
| la27 | 20 × 10 | 0.5 | 1084 | 1085 | 1085 | 1085 | 1084* |
| la28 | 20 × 10 | 0.5 | 1069 | 1070 | 1070 | 1070 | 1069* |
| la29 | 20 × 10 | 0.5 | 993 | 994* | 994* | 994* | 994* |
| la30 | 20 × 10 | 0.5 | 1068 | 1069* | 1069* | 1069* | 1069* |
| la31 | 30 × 10 | 0.5 | 1520 | 1520* | 1520* | 1521 | 1520* |
| la32 | 30 × 10 | 0.5 | 1657 | 1658* | 1658* | 1658* | 1658* |
| la33 | 30 × 10 | 0.5 | 1497 | 1497* | 1497* | 1497* | 1497* |
| la34 | 30 × 10 | 0.5 | 1535 | 1535* | 1535* | 1535* | 1535* |
| la35 | 30 × 10 | 0.5 | 1549 | 1549* | 1549* | 1549* | 1549* |
| la36 | 15 × 15 | 0.5 | 948 | 948* | 948* | 950 | 948* |
| la37 | 15 × 15 | 0.5 | 986 | 986* | 986* | 986* | 986* |
| la38 | 15 × 15 | 0.5 | 943 | 943* | 943* | 943* | 943* |
| la39 | 15 × 15 | 0.5 | 922 | 922* | 922* | 922* | 922* |
| la40 | 15 × 15 | 0.5 | 955 | 955* | 955* | 956 | 955* |

**Note:**
* Best obtained makespan.

**Table 9 Average RPD for the Hurink-rdata instances and Friedman test comparing all the algorithms and the GLNSA with the HA.**

| Algorithm: | GLNSA | HA | IJA | SSPR | $p$ value all algorithms | $p$ value GLNSA vs HA |
|---|---|---|---|---|---|---|
| Average *RPD*: | 0.0772 | 0.0724 | 0.1177 | 0.0593 | 0.0000003 | 0.5637 |
| Rank: | 3 | 2 | 4 | 1 | | |

Nopt1 neighborhood works adequately for problems with greater flexibility. From the results of the four sets of experiments, the GLNSA has obtained results comparable with state-of-art algorithms and with a competitive computational complexity, especially for problems with a high rate of flexibility.

## CONCLUSIONS AND FURTHER WORK

This work has presented an algorithm that performs a global search with smart-cells using a cellular automaton-like neighborhood where individual operators such as insert and swapping are used, along with an operator like PR, to share information between solutions. These operators are primarily focused on optimizing the scheduling of operations.

The local search on the GLNSA performs a TS to find the best assignment of machines for each operation. Another contribution of this work is that a simplified neighborhood based on Nopt1 is proposed, where the feasible machine of a critical operation is modified without explicitly finding the optimal allocation of the operation, since this is left to global search operations, which is suitable for FJSP instances with high rate of flexibility.

The cellular automaton-like neighborhood allows this type of operations to be carried out concurrently and in a balanced way, which provides an equilibrium between the exploration and exploitation of the GLNSA and allows the use of a lower number of smart-cells compared to other algorithms, as well as a lower number of iterations of the TS, which is reflected in a lower computational complexity.

Four well-known benchmarks (including 101 instances) were used to develop the GLNSA's computational experimentation. The results obtained show good performance compared to the algorithms taken as a reference.

The GLNSA represents a new way of solving task scheduling, which can be applied to other types of problems, such as the Flowshop, the Job Shop, or the Open Shop Scheduling Problem, where cellular automaton-like neighborhoods can be applied to make concurrent exploration and exploitation actions.

For possible future work, we propose to use other operations, such as two-point, POX, or JBX crossovers, or other types of mutations, for global search. Other types of local search strategies such as climbing algorithms with restarts can also be used. Also, other simplifications of the Nopt1 neighborhood can be investigated to treat problems with less flexibility.

Finally, the GLNSA approach that uses a cellular automaton-like neighborhood can also be extended to investigate its effectiveness in optimizing multi-objective manufacturing problems.

### Funding
This study was supported by the National Council for Science and Technology (CONACYT) with project number CB- 2017-2018-A1-S-43008, and Nayeli J. Escamilla Serna was supported by CONACYT with grant number 1013175. The funders had no role in study design, data collection and analysis, decision to publish, or preparation of the manuscript.

### Grant Disclosures
The following grant information was disclosed by the authors:
National Council for Science and Technology (CONACYT): CB- 2017-2018-A1-S-43008.
CONACYT: 1013175.

### Competing Interests
The authors declare that they have no competing interests.

### Author Contributions
- Nayeli Jazmin Escamilla Serna performed the experiments, analyzed the data, performed the computation work, prepared figures and/or tables, and approved the final draft.
- Juan Carlos Seck-Tuoh-Mora conceived and designed the experiments, performed the experiments, analyzed the data, performed the computation work, prepared figures and/or tables, authored or reviewed drafts of the paper, and approved the final draft.
- Joselito Medina-Marin analyzed the data, performed the computation work, authored or reviewed drafts of the paper, and approved the final draft.
- Norberto Hernandez-Romero conceived and designed the experiments, performed the experiments, prepared figures and/or tables, and approved the final draft.
- Irving Barragan-Vite conceived and designed the experiments, prepared figures and/or tables, authored or reviewed drafts of the paper, and approved the final draft.
- Jose Ramon Corona Armenta analyzed the data, authored or reviewed drafts of the paper, and approved the final draft.

### Data Availability
The code is available in the Supplemental File, and the code and test problem are also available at GitHub: https://github.com/juanseck/GLNSA-FJSP-2020.

### Supplemental Information
Supplemental information for this article can be found online at http://dx.doi.org/10.7717/peerj-cs.574#supplemental-information.

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
