# Peer review of "A global-local neighborhood search algorithm and tabu search for flexible job shop scheduling problem"

_PeerJ Computer Science, doi:10.7717/peerj-cs.574_

## Round 0.1 · original submission · Major Revisions

Please revise carefully the paper according to the comments of the reviewers.

Reviewer 1 ·

Basic reporting

More recent literature should be included.

The authors should clearly present the novelty and contribution of their algorithm. In my opinion, it seems to be a conventional hybridization of population based and local search methods, which have been addressed adequately in the past.

Experimental design

Benchmark sets are not sufficient. Other existing problem instances should be considered.

Further reference algorithms should be considered.

Validity of the findings

The algorithm proposed actually failed to match existing ones for some benchmark problem instances, given the very limited sets. Moreover, CPU time is not comparable with papers published years ago. These issues all make its performance questionable.

Reviewer 2 ·

Basic reporting

See below

Experimental design

See below

Validity of the findings

See below

Additional comments

-Compare computational results with none-parametric statistical test like Friedman test, Wilcoxon signed-rank test etc. as advised in the literature [J. Derrac, S. Garcia, D. Molina, F. Herrera, A practical tutorial on the use of nonparametric statistical tests as a methodology for comparing evolutionary and swarm intelligence algorithms, Swarm Evol. Comput. 1(1) (2011) 3-18].
-Why tabu search is selected as the main search procedure? More discussions may be useful.
-For mathematical formulation of the problem see [Ozguven et al., Mixed integer goal programming models for the flexible job-shop scheduling problems with separable and non-separable sequence dependent setup times, Applied Mathematical Modelling, Volume 36, Issue 2, February 2012, Pages 846-858]
-There is more recent literature on the problem that need to be considered by replacing older references. See for example
[Huang, X. and Yang, L. (2019), "A hybrid genetic algorithm for multi-objective flexible job shop scheduling problem considering transportation time", International Journal of Intelligent Computing and Cybernetics, Vol. 12 No. 2, pp. 154-174. https://doi.org/10.1108/IJICC-10-2018-0136]
[Baykasoglu, A., Madenoglu, F.S., Hamzadayi, A., Greedy randomized adaptive search for dynamic flexible job-shop scheduling, Journal of Manufacturing Systems, 56, 425-451, 2020].
-Gantt charts are not necessary.

---

## Round 0.2 · accepted · Accept

Authors have provided suitable answers to the comments of the reviewers. The paper is ready to be accepted for publication.